# Tractometry-based Anomaly Detection for Single-subject White Matter Analysis

**Maxime Chamberland**[1]**, Sila Genc**[1]**, Erika P. Raven**[1]**, Greg D. Parker**[1]**, Adam Cunningham**[2]**, Joanne Doherty**[1,2]**, Marianne van den Bree**[2]**, Chantal M. W. Tax**[1]**, Derek K. Jones**[1]

[1] *Cardiff University Brain Research Imaging Centre (CUBRIC), Cardiff, United Kingdom*
[2] *MRC Centre for Neuropsychiatric Genetics and Genomics, Cardiff, United Kingdom*

## Abstract

There is an urgent need for a paradigm shift from group-wise comparisons to individual diagnosis in diffusion MRI (dMRI) to enable the analysis of rare cases and clinically-heterogeneous groups. Deep autoencoders have shown great potential to detect anomalies in neuroimaging data. We present a framework that operates on the manifold of white matter (WM) pathways to learn normative microstructural features, and discriminate those at genetic risk from controls in a paediatric population.

**Keywords:** Diffusion MRI, Tractography, Tractometry, Anomaly Detection, Autoencoder

## 1. Introduction

Considerable effort has gone into designing methods for group comparisons in dMRI (Jones and Cercignani, 2010) (i.e., $N$ patients vs $M$ controls) and as such, single-subject analysis frameworks (i.e., 1 patient vs $M$ controls) are currently lacking. For clinically-heterogeneous groups, normative modeling (Marquand et al., 2016) has been proposed, but often relies on voxel-based methods, and hence is suboptimal for dMRI since WM tracts offer a more suitable manifold. In this work, we investigate individual differences in WM microstructure in children with copy number variants (CNVs) at high genetic risk of neurodevelopmental and psychiatric disorders (Chawner et al., 2019), which are relatively *rare* and challenging to recruit for research imaging studies (Villalón-Reina et al., 2019). We propose the following unsupervised framework for anomaly detection: First, we learn a normative set of features derived from microstructural tract profiles obtained from typically developing (TD) children. Second, we apply the framework to unseen subjects, to determine whether these deviate from controls (based on the hypothesis that deviations will stand out from the normative distribution).

## 2. Methods

### 2.1. Data acquisition & preprocessing

Diffusion data from 90 TD (age 8-18 years) and 8 children with a CNV and no apparent WM lesions (age 8-15 years) were acquired on a Siemens 3T Connectom MRI scanner with 14 b0 images, 30 directions at b = 500, 1200 s/mm$^2$, 60 directions at b = 2400, 4000, 6000

s/mm$^2$ and 2×2×2mm$^3$ voxels. Data were preprocessed as in Chamberland et al. (2019) and rotationally-invariant spherical harmonic (RISH0, Mirzaalian et al. (2015)) features were derived for each subject using the b = 6000 s/mm$^2$ shell. Automated WM tract segmentation was performed using TractSeg (Wasserthal et al., 2018) to obtain 20 bundles of interest (Fig. 1, left). For each bundle, Tractometry (Bells et al., 2011; Cousineau et al., 2017) was performed (sampling at 20 locations, Chamberland et al. (2019)) and the resulting 20 tract profiles were concatenated to form a feature vector (n = 20 tracts × 20 locations = 400 features) for each subject.

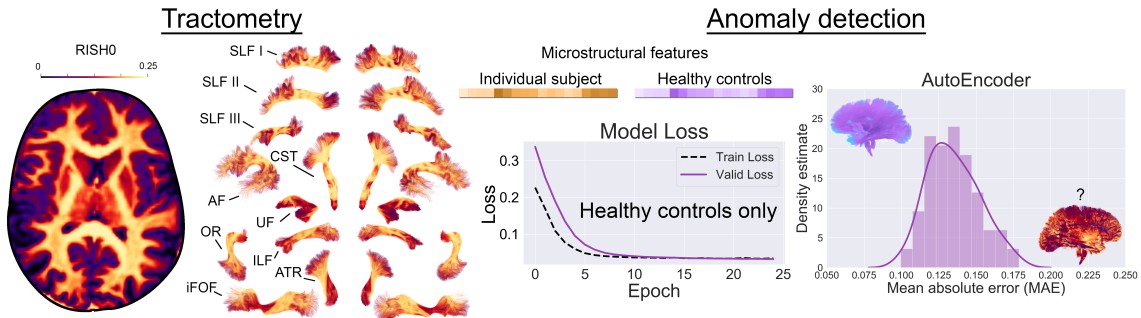

Figure 1: Left: RISH0 feature mapped over 20 WM bundles with Tractometry. Right: The proposed unsupervised anomaly detection framework trained on healthy data only.

## 2.2. Anomaly detection

Our autoencoder architecture consists of five fully connected symmetric layers. The input and output layers have the same number of nodes as the length of the tract features vector. The inner layers consecutively apply a compression ratio of 2 by reducing the number of nodes by half, up to the bottleneck hidden layer (activation: ReLU, epochs: 25, batch size: 24, learning rate: 1.0e-3, optimiser: Adam, loss: mean squared error).

A validation set (n = 16) was generated and held-out by combining the individuals with a CNV (n = 8) with a random subset of TD (n = 8). The rest of the TD (n = 82) data was used to establish a normative distribution (Fig. 1, right). 10% of the TD data was held out for testing during the training phase where the goal was to generate an output ($\hat{x}$) similar to the input ($x$) by minimising the reconstruction error and computing the mean absolute error over all features as anomaly score. Age regression and feature normalization were performed on the training set and subsequently applied to the validation set. We repeated this entire process 50 times to assess variations within the TD population and derived a mean anomaly score for each subject. Using the subject labels, we report the mean ROC area under the curve (AUC) across the iterations. In addition, we compared our results with two previously-reported approaches: 1) univariate z-score distribution (Yeatman et al., 2012); and 2) multivariate PCA combined with the Mahalanobis distance (Yeatman et al., 2018; Sarica et al., 2017; Taylor et al., 2020) using the aforementioned bootstrapped approach.

## 3. Results

The autoencoder approach was better at identifying CNV subjects as outliers (AUC: 0.86 $\pm$ 0.06) compared with z-score (AUC: 0.53 $\pm$ 0.06) and PCA (AUC: 0.61 $\pm$ 0.09). Fig. 2 shows the reconstructed features of a random CNV subject which highlights significant discrepancies along various association pathways, suggesting subject-specific differences in microstructural attributes.

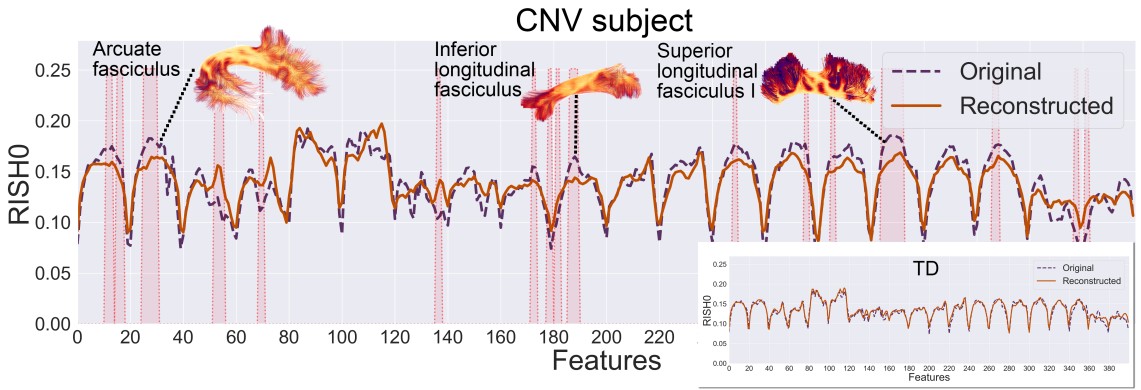

Figure 2: RISH0 features (20 sections $\times$ 20 tracts) of a CNV subject before ($x$) and after ($\hat{x}$) reconstruction. Discrepancies were statistically identified along major association pathways using bootstrapped permutations (shaded area, p $<$ 0.01). In comparison, features from a representative TD show no significant anomalies (bottom right).

## 4. Discussion & future work

The framework enabled *subject-* and *tract-specific* characterisation of WM microstructure. By training on only healthy data, our findings revealed that clinical cases (CNVs) can be classified as outliers. Furthermore, inspection of discrepancies in tract profiles allow the identification of clinically relevant differences in microstructural attributes. The framework also outperformed traditional multivariate outlier detection mostly due to its ability to handle high-dimensional data non-linearly. This extends the possibility of using anomaly detection in clinically-heterogeneous groups (Kia et al., 2018; Wolfers et al., 2020) and extremely rare cases (as little as $N$=1), where group comparisons are otherwise impossible. Future work will include additional cohorts to better assess the generalizability of the framework and its application to other pathology. Importantly, anomalies should be linked to clinical findings (Taylor et al., 2020). Our Tractometry-based anomaly detection framework paves the way to progress from the traditional paradigm of group-based comparison of patients against controls, to a personalised medicine approach, and takes us a step closer in transitioning microstructural MRI from the bench to the beside. The browser-based tool will be made freely-available to the community via Github.

## Acknowledgments

The authors thank Prof. Maxime Descoteaux and Jean-Christophe Houde (Sherbrooke Connectivity Imaging Lab) for their helpful discussions and code sharing. This work was supported by a Wellcome Trust Investigator Award (096646/Z/11/Z), a Wellcome Trust Strategic Award (104943/Z/14/Z), an EPSRC equipment grant (EP/M029778/1) and a Sir Henry Wellcome Fellowship (215944/Z/19/Z).

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
