# OpenReview forum: "Tractometry-based Anomaly Detection for Single-subject White Matter Analysis"
_MIDL.io/2020/Conference — MIDL 2020_

### Official Review · AnonReviewer2 · 2020-03-02
**Interesting work for the diffusion MRI community, but evaluation is not sufficient**

**Rating:** 1
**Confidence:** 4

**Review:**

Summary
The manuscript proposes a method (Autoencoder) for anomaly detection based on Tractometry features.

Quality
The preprocessing of the dMRI data as well as the extraction of the Tractometry features is methodologically sound. Using the reconstruction loss of an Autoencoder to detect anomalies is also a common approach. Moreover, the authors used two other methods (Z-score and PCA) as baseline for their proposed method (Autoencoder). Comparing to more sophisticated models like Variational Autoencoders would have been nice.
My main concern is the following: The test set is extremely small (n=3 out of distribution samples). The authors argue that their method is suitable for single-subjects analysis, however, for a proper evaluation it would have been necessary to have a larger test set. For the 3 out of distribution samples two were correctly detected as out of distribution. This raises the question if the results are significant or only noise. A larger test set is needed to answer this question.
Ideally the authors would also evaluate their proposed method on a second dataset to show that their method generalizes to different datasets (the authors stated this as future work).

Clarity
Not all methodological details are clear, e.g. it is not clear how the PCA+Mahalanobis distance was applied. This at least should be a bit clearer.

Originality and Significance
Using autoencoders for anomaly detection is very well established. But doing this kind of anomaly detection in the field of diffusion MRI and Tractometry is new and certainly valuable for the field. However, a more diffusion MRI focused conference like ISMRM could be a better fit than MIDL which is very Deep Learning focused.

Summary
This paper could be valuable for the diffusion MRI and Tractometry community if the evaluation would be sound. However, using only n=3 out of distribution test samples the evaluation is not really meaningful and can not answer the question if the proposed method really works.

---

### Official Review · AnonReviewer3 · 2020-03-13

**Rating:** 4
**Confidence:** 4

**Review:**

Summary: the paper presents an interesting framework for single-subject analysis. Patients should be seen as 'anomalies' with respect to a normative model. The method is tested on white matter tract profiles with interesting and convincing results (even if the number of observations, especially for patients, is rather small).

Remarks:
1- Authors should better explain what the 20 features per tract represent. Is it as in Cousineau et al. (2017) or as in Chamberland et al. (2019) ? Do they represent average FA in the tract profile ? As authors have mentioned in the conclusions, it would also be interesting to inspect the robustness of the results with respect to this hyper-parameter (number of features per tract). This should be considered indeed as future work.
2- How are the anomaly thresholds chosen in Fig. 1 ? Please clarify.
3- In Fig. 3, authors show the R0 profile of a CNV patient which highlights discrepancies in the association tracts. What about the other patients and controls ? Is this R0 profile an actual outlier with respect to the profiles of the controls ? Please clarify.

---

### Official Review · AnonReviewer1 · 2020-03-13
**Promising preliminary work for anomaly detection using tractometry**

**Rating:** 3
**Confidence:** 4

**Review:**

Authors propose a tractometry-based approach for anomaly detection. The method is based on an autoencoder. Authors use a pretty unique dataset to test their approach. The article is well written. It is organized logically and it is easy to follow.

It is difficult to understand why authors picked such a rare disease case. It is very much contradictory to authors' claim that there is an urgent "need for a paradigm shift for individual diagnosis" and then address a rare disease case to show as preliminary results. Do authors know what kind of changes are expected in these patients' brain? What is the rationale behind the success of the proposed approach? Were there expected differences in certain pathways? How can one be convinced that there is an actual difference? Authors might argue that "that is the point of the study!". However, it is not! These are preliminary results and it is more important to show that this idea can work. Therefore, it is essential to show a case where there are some underlying hypothesis for the disorder, such as depression so readers would know that "aha, they can detect changes in suspicious areas which are hard to find otherwise".

It would also benefit this work if authors study data from more conventional protocols and devices; even though, I would expect that the results would be similar. This however would strengthen the impact of the work to show that it can be applied by a wider range of clinics or research groups.

---

### Official Review · AnonReviewer4 · 2020-03-14
**Unsupervised anomaly detection on images of children with CNV**

**Rating:** 3
**Confidence:** 4

**Review:**

It is a well written short submission. It offers a single-subject analysis of rare cases for diagnostic purposes in contrast with group analysis. The authors present promising results, but the significance at the end is a bit overstated.

The framework offers to learn normative microstructural features via an autoencoder from a TD cohort and uses unsupervised anomaly detection on images of children with CNV (a rare disease). The novelty is on the low side. The image processing of the diffusion images is done using TractSeg. Twenty white matter bundles of interest are reconstructed and Tractometry is run along these, using 20 control points. The performance is compared to classic z-score and PCA analysis. The proposed autoencoder, the novelty of the proposed work, manages to identify 2 outlier subjects, while the other could not.

Minor
======
Both Z-score and Mahalanobis distance thresholds cannot ... --> Neither Z-score nor Mahalanobis distance thresholds can..

---

### Meta-Review · Area_Chair1 · 2020-04-07
**MetaReview of Paper27 by AreaChair1**

**Rating:** 3

**Metareview:**

Although reviewers have raised concerns in terms of novelty and validation, most of them have given a positive evaluation. I therefore recommend acceptance.

**Paper Type:**

both

---

### Decision · Program_Chairs · 2020-04-11

Accept